# Dampened Muscle mTORC1 Response Following Ingestion of High-Quality Plant-Based Protein and Insect Protein Compared to Whey

**DOI:** 10.3390/nu13051396

**Published:** 2021-04-21

**Authors:** Gommaar D’Hulst, Evi Masschelein, Katrien De Bock

**Affiliations:** 1Laboratory of Exercise and Health, Department of Health Sciences and Technology, Swiss Federal Institute of Technology (ETH) Zurich, 8603 Zurich, Switzerland; evi-masschelein@ethz.ch (E.M.); katrien-debock@ethz.ch (K.D.B.); 2Laboratory of Regenerative and Movement Biology, Department of Health Sciences and Technology, Swiss Federal Institute of Technology (ETH) Zurich, 8093 Zurich, Switzerland

**Keywords:** muscle, mTORC1, plant-based protein, whey, insect, muscle protein synthesis

## Abstract

Increased amino acid availability acutely stimulates protein synthesis partially via activation of mechanistic target of rapamycin complex 1 (mTORC1). Plant-and insect-based protein sources matched for total protein and/or leucine to animal proteins induce a lower postprandial rise in amino acids, but their effects on mTOR activation in muscle are unknown. C57BL/6J mice were gavaged with different protein solutions: whey, a pea–rice protein mix matched for total protein or leucine content to whey, worm protein matched for total protein, or saline. Blood was drawn 30, 60, 105 and 150 min after gavage and muscle samples were harvested 60 min and 150 min after gavage to measure key components of the mTORC1 pathway. Ingestion of plant-based proteins induced a lower rise in blood leucine compared to whey, which coincided with a dampened mTORC1 activation, both acutely and 150 min after administration. Matching total leucine content to whey did not rescue the reduced rise in plasma amino acids, nor the lower increase in mTORC1 compared to whey. Insect protein elicits a similar activation of downstream mTORC1 kinases as plant-based proteins, despite lower postprandial aminoacidemia. The mTORC1 response following ingestion of high-quality plant-based and insect proteins is dampened compared to whey in mouse skeletal muscle.

## 1. Introduction

Maintenance of skeletal muscle mass is key for metabolic health throughout life [1]. Total muscle mass is generally determined by basal rates of protein synthesis and the ability to stimulate protein synthesis after the intake of protein [2,3]. The enhancement of protein synthesis is primarily induced by the activation of mechanistic target of rapamycin complex 1 (mTORC1)-signalling [4,5,6,7]. Growth factors, energetic stress, and muscular contractions control mTORC1 directly or via its upstream inhibitor tuberous sclerosis complex 1/2 (TSC1/2) [8]. Amino acids (AA) regulate mTORC1 via alternative pathways involving Rag GTPases, which recruit mTORC1 to the lysosomal membrane [9,10,11]. How TSC2-dependent signals regulate mTORC1 in skeletal muscle has been intensely studied, but how fluctuations of amino acids, and in particular fluctuations in the essential amino acid leucine, regulate mTORC1, and thus skeletal muscle mass in vivo is much less well understood.

Our world’s population is projected to reach 10 billion by 2050 [12] and protein demand is expected to double by then [13]. In the future, we will no longer be able to cover the animal-based protein demand at rates we are consuming now [14]. Plant-based proteins currently account for 30–50% of total dietary protein intake in many countries [15], with numbers exceeding 60% in less developed countries [16]. More research is emerging on how plant-based proteins affect muscle health [17,18,19] and muscle remodelling, especially in combination with resistance training [20,21,22,23]. Considerable efforts have been made recently to understand how non-animal-based proteins affect the acute protein synthetic response [24,25,26,27], but studies assessing the direct effects on acute mTORC1 signalling are lacking.

A rapid postprandial rise in essential amino acids (EAA) concentrations modulates the increase in mTORC1 activity [28,29] after protein ingestion. As animal-based protein sources are rapidly digested and contain high concentrations of EAA, in particular leucine, it is believed that animal-based proteins are more anabolic than plant-based sources [17]. The latter contain several anutritious molecules such as tannins, trypsin inhibitors, and phytates that slow digestion [30]. Moreover, plant-based proteins result in larger portions of splanchnic nitrogen retention/oxidation and hence lower AA appearance in the blood [31]. Nevertheless, innovations in food processing have solved many issues regarding plant-based protein digestibility. Recent advancements in production of plant-based protein concentrates, isolates, and hydrolysates and the potential of mixing different plant sources have been put forward as a strategy to match EAA content of animal-derived protein sources [22], but these hypotheses are yet to be tested. In addition to plant-based proteins, insect-based protein mixtures might be another promising strategy to meet the rapidly growing protein demand. Insect protein has been reported to induce a similar postprandial increase in essential AA (EAA) as soy, but inferior to whey [32]. Similar to vegan proteins, the effects of insect protein isolates on mTORC1 activity remain unexplored.

Hence, the aim of this study is to assess AA appearance in the blood and subsequent mTORC1 signalling acutely after administration of whey or of more sustainable, high quality protein sources based on plants (pea and rice isolate) and insects (buffalo worm protein). We hypothesise that whey protein would promote superior blood concentrations of EAA and branched-chain amino acids (BCAA) compared to total protein-matched vegan proteins or insect isolates, mirrored by superior acute mTORC1 activation. Additionally, we hypothesise to observe a similar EAA and mTORC1 response to whey when a vegan protein mix is leucine-matched to whey.

## 2. Materials and Methods

### 2.1. Animals

All experiments were performed on male C57BL/6J mice. Mice were housed in individually ventilated cages (3–4 littermates per cage) at standard housing conditions (22 °C, 12 h light/dark cycle, dark phase starting at 7 pm), with ad libitum access to chow (KlibaNafag, diet #3436 and diet #3437) and water. Health status of all mouse lines was regularly monitored according to FELASA guidelines. All animal procedures were approved by the Veterinary office of the Canton of Zürich (license nr ZH137/2020).

### 2.2. Experimental Procedures

An overview of the experimental procedures can be found in Figure 1. A number of 8–12-week old male C57BL/6J mice were fasted for 4–5 h from the beginning of the light cycle. Subsequently, mice were gavaged with saline (0.9% NaCl), Whey (myprotein, Northwitch, UK), a pea–rice Vegan Mix (UniProt, Kaltenkirchen, Germany), Vegan Mix++ leucine-matched to whey (Uniprot) and pulverized buffalo Worm protein (Protifarm, Ermelo, The Netherlands) (Table 1). Blood was drawn 30, 60, 105 and 150 min after gavage, mice were sacrificed 60 min and 150 min after gavage for harvesting muscle samples. Each AA-containing condition contained the same amount of total protein per mixture (4.57 g·kg^−1^), except Vegan Mix++ (5.48 g·kg^−1^). The Vegan Mix++ was leucine matched to whey (0.475 g leucine·kg^−1^), hence it contained 20% more total protein. The concentrations chosen correspond to ~40% of total daily leucine intake and have been shown to induce a robust increase in mTORC1 signaling and MPS in rodents [29,33]. AA content of each amino acid mixture was internally assessed by the respective company where the protein source was obtained. Nitrogen to protein ratio (N:P) for whey, vegan mix and worm protein is 6.38, 5.17–5.44 and 5.41, respectively. Values are obtained from previous studies [34,35,36,37]. An overview of AA mixture specifications is provided Table 1.

### 2.3. Sample Collection

Mice were anesthetized using Ketamine/Xylazine at 115 μg·g^−1^ and 13 μg·g^−1^ body weight respectively via intraperitoneal injection. The depth of anaesthesia was confirmed by testing pedal withdrawal reflex before tissue collection. Subsequently, the m. gastrocnemius (GAS), m. tibialis anterior (TA), m. soleus (SOL), m. plantaris (PLT) were dissected and snap frozen.

### 2.4. Protein Extraction and Western Blot

Sample Preparation: Between 10 and 25 mg of muscle sample (m. tibialis anterior, TA) was homogenized in ice cold lysis buffer (1:10, *w/v*) (50 mM Tris-HCl pH 7.0, 270 mM sucrose, 5 mM EGTA, 1 mM EDTA, 1 mM sodium orthovanadate, 50 mM glycerophosphate, 5 mM sodium pyrophosphate, 50 mM sodium fluoride, 1 mM DTT, 0.1% Triton-X 100 and 10% protease inhibitor) (20 μL per 1.8–2.5 g of tissue sample) using an OMNI-THq Tissue homogenizer (OMNI International, Kennesaw, GA, USA) for 20 s until a consistent homogenate was formed. Samples were centrifuged at 4 °C at 10,000× *g* for 10 min and the supernatant with proteins collected. Protein concentration was determined using the DC assay protein method to equalize the amount of protein. Samples were prepared 3:4 with 4× laemmli buffer containing 10% 2-mercaptoethanol and heated at 95° for 5 min. An amount of 20–40 µg of total protein was loaded in a 15-well pre-casted gradient gel (Bio-rad, #456–8086). After electrophoresis, a picture of the gel was taken under UV-light to determine protein loading using stain-free technology. Proteins were transferred via semi-dry transfer onto a polyvinylidene fluoride membrane (Bio-rad, #170–4156) and subsequently blocked for 1 h at room temperature with 5% milk in TBS- Tween. Membranes were incubated overnight at 4 °C with primary antibodies from Cell Signaling Technology; pS6K1^Thr389^ (#9206), pS6^Ser235/236^ (#2211), pmTOR^Ser2448^ (#5536), peEF2^Thr56^ (#2331) and p4E-BP1^Ser65^ (#9451). The appropriate secondary antibodies (1:5000) for anti-rabbit and anti-mouse IgG HRP-linked antibodies (Cell signalling, #7074) were used for chemiluminescent detection of proteins. Membranes were scanned with a chemidoc imaging system (Bio-rad) and quantified using Image lab software (Bio-rad).

### 2.5. Amino Acid Determination from Blood

A total of 5 nmol of stable isotope labelled amino acids was added (Cambridge Isotope Laboratories) to 10 µL of serum. Proteins were precipitated by adding 9 volumes ice cold methanol at −20 °C. The supernatant was dried and the amino acids were re-constituted in 100 µL 0.1 % acetic acid. Samples (5 µL) were subjected to liquid chromatography coupled multiple reaction monitoring mass spectrometry (LC-MRM-MS). The acquired data were integrated and analysed using the open source software tool Skyline [38]. Area under the Curve (AUC) was calculated using the linear trapezoidal method.

### 2.6. Statistical and Data Analyses

Results are presented as mean with standard error of the mean (SEM) bars. Additional individual data points are provided for the protein signalling data. Data were subjected to a one-way analysis of variance (ANOVA) (protein signalling) or two-way ANOVA (plasma AA) to generate a *p* value and post hoc tests were performed using Tukey’s post hoc test using Graphpad Prism to compare between groups. All data passed the Shapiro–Wilk test for normality and sample sizes (*n* = 5–7) were calculated a priori via power calculations (1-β: 0.8) using G*Power statistical software. Significance was set at *p* < 0.05. Exact *n* numbers are provided in the figure and table legends.

## 3. Results

### 3.1. Amino Acid Concentrations in Plasma

The postprandial rise in EAA, and in particular BCAA, is essential for mTORC1 activation and subsequent induction of muscle protein synthesis (MPS) in skeletal muscle. BCAA appearance in blood plasma is shown in Figure 2 and Table 2. Plasma leucine increased 30, 60 and 105 min after whey gavage (Figure 2A) and returned to baseline levels after 150 min. Vegan Mix and Vegan Mix++ induced a dampened plasma leucine response compared to whey, with no differences between conditions. Worm protein induced the lowest leucine response, which was lower in comparison to both Vegan Mix, Vegan Mix++ as well as whey (Figure 2A and Table 2). All this resulted in a reduced AUC for both plant-based mixtures and worm protein compared to whey (Figure 2B). The other two BCAA, isoleucine and valine, followed near identical patterns (Figure 2C–F). Essential amino acids (AA) methionine and threonine also displayed a higher, more prolonged peak after whey supplementation compared to Vegan Mix, Vegan Mix++ and worm protein (Table 2). Other amino acids were only marginally affected by any of the AA mixtures (for details, see Table 2).

### 3.2. Higher Acute mTORC1 Response with Whey Compared to Plant- and Insect-Based Protein

Blood analysis showed that BCAAs peaked 30–60 min after gavage. Hence, to evaluate whether the higher BCAA response 60 min after whey supplementation would also lead to a higher mTORC1 response in skeletal muscle, we measured the phosphorylation of critical downstream mTORC1 targets ribosomal protein S6 kinase (pS6K1^Thr389^), ribosomal protein S6 (pS6^ser235/236^) and eukaryotic elongation factor 2 (peEF2^Thr56^) exactly 60 min after gavage. All AA mixtures strongly increased pS6K1 and pS6, but the effect was larger with whey (∼8-fold vs. saline) compared to plant-based and insect-based protein (∼5-fold), with no differences between both vegan mix conditions and worm (Figure 3A–C,E). Phosphorylation of mTOR at Ser2448 showed an identical pattern (Figure 3C,E). peEF2 remained unaffected by supplementation (Figure 3D,E).

### 3.3. No Prolonged mTORC1 Response with Plant- or Insect-Based Protein

Previous reports in humans have shown a delayed but overall elevated muscle synthetic response with a plant-based source compared to whey, despite a lower overall postprandial amino acid availability with the former [24]. Hence, to evaluate the potential of a delayed mTORC1 response with animal- and/or insect-based protein sources, we assessed critical components of mTORC1 2.5 h after gavage. pS6K1 and pS6 where marginally increased in all AA conditions, but this did not reach statistical significance due to higher variation compared to the acute (60 min after gavage) condition (Figure 4A,B,E). Phosphorylation of 4E-BP1 at Ser65 was increased three-fold in the Vegan mix++ condition (Figure 4C,E, *p* < 0.05). Phospho-mTOR at Ser2448 remained unaffected in all conditions (Figure 4D,E).

## 4. Discussion

This study assessed changes in AA concentrations in the blood and mTORC1 response after gavage of animal-based proteins (whey), plant-based protein (a pea–rice vegan mix) and insect proteins (pulverized worm). The major findings were that all protein sources robustly increased aminoacidemia and subsequently mTORC1 signalling in mouse skeletal muscle acutely after gavage. However, whey induced a ∼2-fold larger increase in plasma BCAA compared to plant-and insect-based proteins, with a subsequent 1.5-fold increase in downstream mTORC1 signalling. A plant-based protein source that was leucine-matched to whey did not rescue the dampened mTORC1 response. Finally, 2.5 h after gavage, aminoacidemia returned to baseline in all conditions, with no obvious prolonged mTORC1 response with plant-based, nor insect-based proteins.

Ingestion of plant-based proteins results in a lower postprandial muscle synthetic response compared to nitrogen-matched animal-based sources such as whey [26,27], skimmed milk [27] or beef [39] in both resting and post-exercise conditions. Lower anabolic properties of single source plant-based proteins may be attributed to inferior BCAA, in particular, leucine, concentrations compared to most animal proteins [17]. Mixing different plant strains to obtain an amino acid profile similar to those of animal sources might be a promising strategy to enhance protein quality of plant proteins [40,41,42]. Yet, our data does not confirm this, as the high-quality pea–rice vegan mix used in this study caused a strongly reduced peak in leucine uptake in the blood compared to whey, even when total leucine content was matched (Vegan Mix++). Increased splanchnic extraction/oxidation and subsequent urea synthesis have been shown with plant-based proteins [31] as well as reduced digestion due to the presence of anutritious components such as tannins, trypsin inhibitors and phytates [30]. However, the latter is unlikely an important factor in our study set-up as both vegan mixtures lead to an overall lower AUC for leucine compared to whey. In effect, a delay in AA absorption would have resulted in a similar overall AUC. Interestingly, although the leucine-matched vegan mix (Vegan Mix++) contained 20% more total protein and 25% more leucine than the normal vegan mix, AA appearance was identical, suggesting high splanchnic retention with higher doses of plant-based proteins.

Next to plant-based protein, other eco-friendly protein options, such as insect proteins, are currently investigated in terms of quality and downstream effects on muscle metabolism. Our data is in line with previous human results showing that insect protein induces blood AA concentrations similar to soy and inferior to whey protein [32]. In fact, we demonstrate that worm protein resulted in a ∼30% lower AUC for all BCAA compared to both vegan mixes. It is important to note that although total administered protein was identical to whey and the pea–rice vegan mix, total leucine content was, respectively 38% and 21% lower, which might explain the slightly reduced BCAA blood appearance with insect protein. Similarly, as shown in Table 1, the worm protein contains more carbohydrates and fats than both the vegan mix and whey. The presence of other macronutrients, in particular fats, might have delayed gastric emptying and AA absorption in our study [34]. Future studies should investigate whether an insect source matched for leucine to whey and stripped from excess fats induces a similar leucine response.

BCAAs and in particular leucine trigger protein synthesis via the activation of mTORC1 [4,28,43,44]. Numerous studies have shown potent activation of downstream mTORC1 signalling with whole animal-based protein sources [28,45,46], but data on mTORC1 activation with non-animal-based protein blends is scarce. Recent human data showed a reduced increase in blood leucine with ingestion of fungal derived protein (mycoprotein) compared to milk protein, even though total ingested leucine was matched [23]. Despite lower leucine availability, MPS was higher with mycoprotein in both resting and exercised conditions, while mTOR phosphorylation at ser2448 was unaffected. As no other downstream kinases were assed in this study, it is difficult to draw strong conclusions on how reduced leucine availability in the mycoprotein group affected mTORC1 signalling. Earlier reports have shown a larger increase in *p*-S6K1 in rats fed a meal containing 20% whey compared to 20% soy [47], which is in line with reduced peak in plasma leucine after consumption of various plant-based proteins of which leucine content was matched to whey [42]. Our data supports this as both vegan mixes induced lower rise in BCAA, which resulted in a 1.5-fold lower increase in phosphorylation of various downstream mTORC1 kinases such as *p*-S6K1 and *p*-S6 compared to whey. Plant and insect protein are often described as slow-digesting [32,48] leading to a prolonged increase in protein synthetic rate, despite lower overall postprandial amino acid concentrations, also at later time-points [23,24]. Although we only measured markers of protein synthesis, we did not observe a prolonged mTORC1 response 2.5 h after plant-or insect protein administration, suggesting that mTORC1 activation closely mirrors the rise in plasma leucine, while its downstream effects on protein synthesis might be delayed. Furthermore, the fact that acute (after 60 min) and prolonged (after 2.5 h) mTORC1 activation was similar between the vegan mixes and insect proteins, despite a lower aminoacidemia in the latter, suggests that the origin of ingested proteins can alter mTORC1′s sensitivity to leucine, possibly via other micro- or macronutrients [23].

The clinical relevance of our data lies in the nutritional support for alternative protein sources to activate mTORC1 and potentially muscle remodeling. Although the doses used in this study are similar to what is generally used in human studies after accounting for dose translation from rodent to humans [49], caution must be exercised to directly extrapolate our findings to humans. Mice and humans show differential digestion patterns. While mice have an exponential digestion curve, where most of the gastric emptying is finished within the first hour after nutrient ingestion, humans show a more linear digestion pattern [50]. This is a limitation of our study as interspecies digestion patters might have affected blood leucine concentrations at later time-points (e.g., 150 min) after gavage.

## 5. Conclusions

In conclusion, ingestion of plant-based proteins matched for total protein and leucine induce a lower rise in blood leucine, with a subsequent dampened mTORC1 activation, compared to whey, both acutely and 2.5 h after administration. Insect protein elicits a similar activation of downstream mTORC1 kinases as plant-based proteins, despite lower postprandial aminoacidemia.

Therefore, future research should focus on the mechanisms behind altered mTORC1 sensitivity related to different protein sources. Finally, further methods to increase the ability of plant-based proteins to activate muscle protein synthesis should be explored; these might include improved means to increase digestion and absorption.

## Figures and Tables

**Figure 1 nutrients-13-01396-f001:**
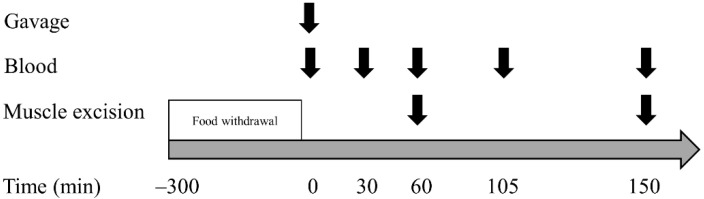
Experimental overview.

**Figure 2 nutrients-13-01396-f002:**
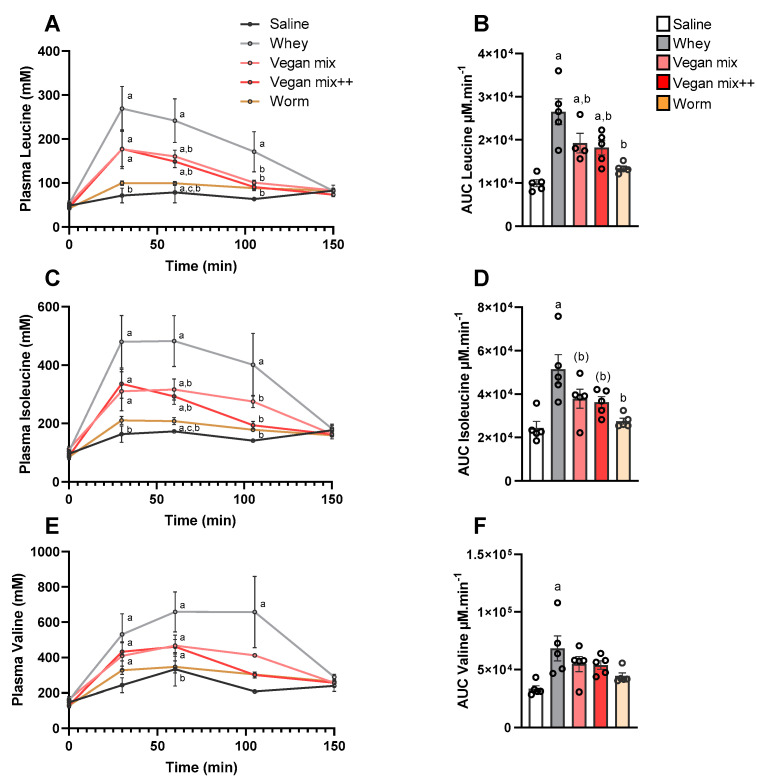
Increase in plasma BCAA upon gavage of different AA mixtures. (**A**) Plasma leucine. (**B**) Leucine area under the curve (AUC). (**C**) Plasma isoleucine. (**D**) Isoleucine AUC. (**E**) Plasma Valine. (**F**) Valine AUC. Data is presented as mean ± SEM, *n* = 5. ^a^
*p* < 0.05 vs. saline, ^b^
*p* < 0.05 vs. whey, ^(b)^
*p* < 0.10 vs. whey. Vegan mix ^++^ annotates the leucine-matched (to whey) vegan mix.

**Figure 3 nutrients-13-01396-f003:**
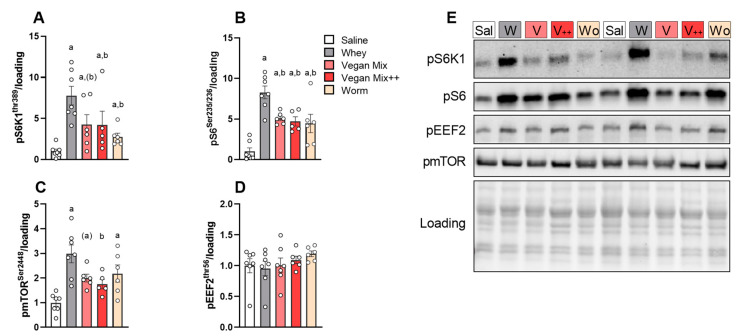
mTORC1 signalling 60 min in *m. Tibialis Anterior* after gavage of different AA mixtures. (**A**) pS6K1^Thr389^. (**B**) pRPS6^Ser235/236^. (**C**) pmTOR^Ser2448^. (**D**) pEEF2^Thr56^. (**E**) Representative blots. Data is presented as mean ± SEM, *n* = 6–7. Sal (Saline), W (Whey), V (Vegan Mix), V++ (Vegan Mix++), Wo (Worm protein). ^a^
*p* < 0.05 vs. saline, ^b^
*p* < 0.05 vs. whey, ^(a)^
*p* < 0.10 vs. saline, ^(b)^
*p* < 0.10 vs. whey.

**Figure 4 nutrients-13-01396-f004:**
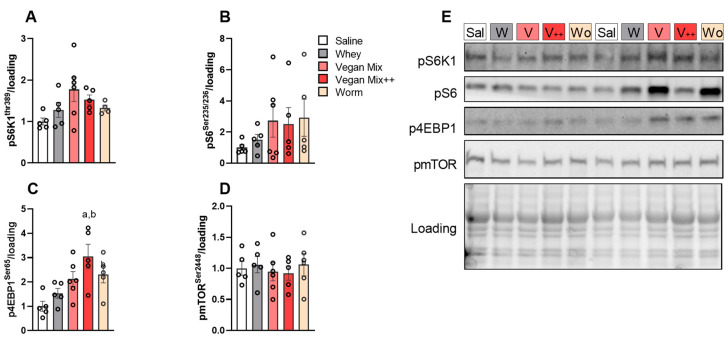
mTORC1 signalling 150 min in *m. Tibialis Anterior* after gavage of different AA mixtures. (**A**) pS6K1Thr389. (**B**) pRPS6Ser235/236. (**C**) p4EBP1Ser65 (**D**) pmTORSer2448. (**E**) Representative blots. Valine AUC. Data is presented as mean ± SEM, *n* = 5–6. Sal (Saline), W (Whey), V (Vegan Mix), V++ (Vegan Mix++), Wo (Worm). ^a^
*p* < 0.05 vs. saline, ^b^
*p* < 0.05 vs. whey.

**Table 1 nutrients-13-01396-t001:** Amino Acid (AA) mixture specifications.

Composition (g/100g Mix)	Whey		Vegan Mix		Worm Protein	
Protein	80.0		78.0		61.7	
Fat	6.7		0.9		24.3	
Carbohydrate	5.0		3.2		8.0	
AA profile (g/100g)	100 g AA	100 g mix	100 g AA	100 g mix	100 g AA	100 g mix
Lysine	9.24	7.39	6.60	5.15	7.08	4.05
Histidine	1.89	1.51	2.40	1.87	3.58	2.05
Methionine	2.01	1.61	1.60	1.25	1.50	0.86
Phenylanine	3.10	2.48	5.30	4.13	4.72	2.70
Threonine	6.68	5.34	3.40	2.65	4.37	2.50
leucine	10.43	8.34	8.30	6.47	7.20	4.12
Isoleucine	6.31	5.05	4.30	3.35	4.61	2.64
Valine	5.80	4.64	5.30	4.13	6.08	3.48
Alanine	4.93	3.94	4.40	3.43	7.08	4.05
Arginine	2.38	1.90	8.60	6.71	5.80	3.32
Aspartic Acid	10.74	8.59	11.20	8.74	9.67	5.53
Glycine	1.75	1.40	4.10	3.20	4.95	2.83
Glutamic Acid	17.73	14.18	19.10	14.90	12.50	7.15
Cystine	1.87	1.50	1.80	1.40	0.94	0.54
Proline	6.27	5.02	4.00	3.12	6.33	3.62
Serine	4.55	3.64	5.00	3.90	4.53	2.59
Tyrosine	2.88	2.30	3.60	2.81	7.80	4.46
Tryptophan	1.45	1.16	1.10	0.86	1.26	0.72
Amount gavaged (g·kg^−1^)	Saline	Whey	Vegan Mix	Vegan Mix++	Worm	
Total protein	0	4.57	4.57	5.48	4.57	
Leucine	0	0.48	0.38	0.48	0.30	

**Table 2 nutrients-13-01396-t002:** Amino acid (AA) in blood plasma after administration of different AA-mixtures.

	0 min					30 min					60 min					105 min					150 min				
	Saline	Whey	Vegan Mix	Vegan Mix++	Worm	Saline	Whey	Vegan Mix	Vegan Mix++	Worm	Saline	Whey	Vegan Mix	Vegan Mix++	Worm	Saline	Whey	Vegan Mix	Vegan Mix++	Worm	Saline	Whey	Vegan Mix	Vegan Mix++	Worm
Essential AA (µM) *n* = 5																									
Lysine	229.4 ± 22.5	253.2 ± 15.5	267.4 ± 20.2	211.7 ± 19.1	216.9 ± 15.9	396.5 ± 65.3	840.6 ± 167.3 ^a^	635.8 ± 122.0 ^a^	643.1 ± 88.5 ^a^	477.8 ± 45.2 ^bd^	505.7 ± 128.6	938.9 ± 201.5 ^a^	664.9 ± 70.3 ^a^	663.0 ± 61.8 ^a^	446.0 ± 41.8 ^bcd^	301.4 ± 45.2	1014.1 ± 452.1	580.2 ± 7.1	410.1 ± 34.0	435.2 ± 50.1	304.5 ± 42.2	399.4 ± 73.5	332.9 ± 25.3	294.9 ± 25.1	308.9 ± 18.0
Histidine	47.9 ± 4.6	59.2 ± 5.6	69.0 ± 5.3	56.3 ± 4.7	51.6 ± 3.3	65.5 ± 5.0	75.5 ± 12.5	78.2 ± 4.2	80.4 ± 10.7	74.9 ± 4.2	94.8 ± 17.0	95.7 ± 17.4	98.9 ± 2.4	96.0 ± 13.5	79.0 ± 3.5 ^c^	72.0 ± 6.0	125.3 ± 18.6	90.1 ± 8.0	76.8 ± 8.1	84.4 ± 9.0	84.1 ± 14.1	73.4 ± 4.3	64.3 ± 4.8	66.2 ± 4.8	65.5 ± 1.9
Methionine	56.7 ± 7.7	53.7 ± 3.7	61.7 ± 7.0	46.8 ± 4.8	44.6 ± 6.7	64.1 ± 3.1	124.4 ± 29.1 ^a^	83.9 ± 15.2 ^b^	78.8 ± 15.3 ^b^	71.9 ± 5.6	64.0 ± 1.9	128.8 ± 32.0 ^a^	66.3 ± 10.1 ^b^	56.9 ± 8.9 ^b^	70.1 ± 6.0 ^b^	44.5 ± 5.0	119.3 ± 30.0 ^a^	62.1 ± 3.7 ^b^	44.4 ± 0.2 ^b^	56.2 ± 6.0 b	39.4 ± 1.9	39.9 ± 2.6	34.3 ± 2.1	31.5 ± 1.2	37.4 ± 2.0
Phenyalanine	114.7 ± 6.7	139.8 ± 4.2	132.7 ± 16.2	116.8 ± 9.9	115.1 ± 6.1	173.4 ± 15.2	231.4 ± 28.2 ^a^	231.0 ± 13.4 ^a^	246.5 ± 7.8 ^a^	214.1 ± 10.2 ^ad^	193.1 ± 9.0	226.6 ± 30.7	241.3 ± 9.9 ^a^	259.8 ± 11.5 ^a^	217.1 ± 5.9 ^cd^	178.7 ± 2.0	245.8 ± 83.6	234.1 ± 8.8	210.6 ± 4.9	226.6 ± 7.8	183.9 ± 12.0	170.1 ± 12.9	180.6 ± 8.0	170.7 ± 5.2	178.1 ± 12.1
Threonine	129.3 ± 19.7	136.3 ± 9.5	143.5 ± 22.3	115.4 ± 14.2	118.8 ± 13.2	183.3 ± 8.7	376.3 ± 83.2 ^a^	287.8 ± 53.1 ^a^	268.3 ± 49.1 ^a^	217.4 ± 22.4 ^b^	232.0 ± 29.3	455.9 ± 96.4 ^a^	294.1 ± 22.9	266.9 ± 29.2 ^b^	215.5 ± 22.4 ^bc^	145.0 ± 19.5	479.1 ± 176.2	259.8 ± 14.6	187.7 ± 7.6	192.3 ± 8.0	143.6 ± 19.1	157.7 ± 22.8	132.3 ± 5.4	137.1 ± 15.9	122.9 ± 9.4
Leucine	48.4 ± 6.5	53.1 ± 1.4	52.4 ± 6.4	44.7 ± 4.9	42.4 ± 3.3 ^b^	71.6 ± 16.7	269.2 ± 50.5 ^a^	177.0 ± 44.5 ^a^	177.5 ± 40.0 ^a^	99.9 ± 4.9 ^b^	78.5 ± 23.7	241.8 ± 49.5 ^a^	160.6 ± 14.2 ^ab^	148.8 ± 13.9 ^ab^	99.3 ± 4.5 ^bcd^	63.5 ± 15.5	171.2 ± 46.5 ^a^	100.9 ± 5.5 ^b^	90.8 ± 8.0 ^b^	88.2 ± 4.9 ^b^	82.6 ± 12.8	81.7 ± 5.8	83.4 ± 5.8	73.5 ± 4.4	82.8 ± 5.3
Isoleucine	96.5 ± 10.4	104.8 ± 2.6	112.3 ± 9.3	88.6 ± 9.3	85.1 ± 9.3 ^b^	163.3 ± 28.1	479.7 ± 89.9 ^a^	310.3 ± 67.1 ^a^	335.9 ± 49.4 ^a^	210.4 ± 5.1 ^b^	172.9 ± 87.0	482.1 ± 36.2 ^a^	316.3 ± 27.9 ^ab^	293.0 ± 12.3 ^ab^	208.0 ± 10.9 ^bcd^	141.2 ± 15.2	401.4 ± 107.7 ^a^	275.4 ± 21.2 ^b^	194.0 ± 13.3 ^b^	178.2 ± 15.5 ^b^	176.6 ± 20.4	182.2 ± 9.8	163.8 ± 9.8	162.5 ± 6.3	159.2 ± 13.45
Valine	147.1 ± 16.5	159.3 ± 5.8	164.6 ± 14.7	130.2 ± 13.2	128.5 ± 10.3	243.5 ± 42.1 ^b^	531.2 ± 116 ^a^	409.9 ± 79.5 ^a^	433.1 ± 51.5 ^a^	328.2 ± 23.7	333.8 ± 94.7	658.7 ± 113 ^a^	467.3 ± 60.6 ^a^	461.4 ± 41.4 ^a^	347.3 ± 34.3 ^b^	208.2 ± 50.1	657.9 ± 202 ^a^	411.7 ± 4.8	300.8 ± 16.8	304.0 ± 20.3	239.9 ± 32.4	289.9 ± 16.2	256.3 ± 21.0	257.5 ± 6.3	262.6 ± 13.4
Non-essential AA (µM)																									
Alanine	205. ± 23.4	213.2 ± 15.6	228.8 ± 29.0	170.0 ± 18.8	197.6 ± 18.6	263.8 ± 19.7	364.7 ± 73.9	350.1 ± 35.2	336.9 ± 43.3	305.9 ± 21.0	372.5 ± 59.3	412.6 ± 71.8	374.1 ± 18.1	389.0 ± 48.3	325.9 ± 21.9	221.7 ± 20.3	420.3 ± 170.3	357.3 ± 67.5	287.2 ± 22.6	306.3 ± 19.2	219.8 ± 29.5	204.6 ± 15.3	199.0 ± 4.5	228.7 ± 28.5	223.0 ± 20.3
Arginine	85.9 ± 10.4	90.9 ± 7.2	94.6 ± 7.0	69.4 ± 5.3	84.8 ± 4.5	106.4 ± 29.8	144.4 ± 21.9	172.3 ± 29.4 ^a^	168.2 ± 7.8	127.7 ± 15.1 ^d^	113.2 ± 26.1	155.1 ± 37.3	152.2 ± 26.2	176.3 ± 25.9	109.5 ± 8.2 ^d^	48.7 ± 16.2	121.2 ± 23.5	157.8 ± 39.4	135.5 ± 29.8	104.8 ± 10.2	46.9 ± 14.2	25.9 ± 7.9	35.2 ± 8.2	34.0 ± 7.5	24.4 ± 4.7
Aspartate	70.2 ± 14.7	81.0 ± 10.8	82.7 ± 15.1	72.2 ± 8.6	78.9 ± 20.7	68.0 ± 11.5	130.0 ± 41.6 ^a^	84.1 ± 15.0	106.7 ± 18.6	65.4 ± 11.2	93.4 ± 33.8	133.7 ± 35.8 ^a^	95.4 ± 8.1	106.6 ± 18.9	81.4 ± 13.3	52.3 ± 25.6	128.2 ± 35.5 ^a^	85.5 ± 16.4	97.7 ± 15.9	76.8 ± 17.9	36.3 ± 14.8	23.1 ± 8.9	19.1 ± 4.2	15.2 ± 4.1	17.6 ± 2.0
Cysteine	1.1 ± 0.4	0.5 ± 0.2	0.7 ± 0.3	0.4 ± 0.2	0.9 ± 0.5	2.4 ± 1.4 ^a^	1.3 ± 0.6	1.0 ± 0.5	1.3 ± 0.3	2.2 ± 1.1	1.4 ± 0.6	4.9 ± 3.3 ^a^	1.7 ± 0.4	1.5 ± 0.3	1.5 ± 0.3	3.0 ± 0.1	6.3 ± 3.8	1.5 ± 0.3	1.5 ± 0.8	1.4 ± 0.5	2.3 ± 0.5	2.0 ± 0.3	3.0 ± 1.1	2.1 ± 0.3	3.3 ± 0.8
Glutamine	84.2 ± 12.6	98.0 ± 19.2	95.8 ± 10.7	88.5 ± 7.9	71.8 ± 11.7	75.8 ± 11.3	159.1 ± 75.2 ^a^	125.8 ± 21.8 ^a^	126.6 ± 16.0 ^a^	88.9 ± 11.2	137.2 ± 42.0	201.3 ± 67.9 ^a^	130.2 ± 11.7 ^b^	162.9 ± 30.4 ^b^	113.2 ± 32.3	64.2 ± 18.3	225.1 ± 75.6	189.3 ± 71.0	181.3 ± 42.1	142.0 ± 23.6	44.6 ± 11.0	25.6 ± 2.0	27.7 ± 3.1	27.6 ± 4.6	28.9 ± 3.7
Glycine	1.1 ± 0.4	0.5 ± 0.2	0.7 ± 0.3	0.4 ± 0.2	0.9 ± 0.5	2.4 ± 1.4 ^a^	1.3 ± 0.6	1.0 ± 0.5	1.3 ± 0.3	2.2 ± 1.1	1.4 ± 0.6	4.9 ± 3.3 ^a^	1.7 ± 0.4	1.5 ± 0.3	1.5 ± 0.3	3.0 ± 0.2	6.3 ± 3.8	1.5 ± 0.3	1.5 ± 0.8	1.4 ± 0.5	2.3 ± 0.5	2.0 ± 0.3	3.0 ± 1.0	2.1 ± 0.3	3.3 ± 0.3
Proline	71.2 ± 6.6	76.4 ± 2.9	83.1 ± 12.5	62.6 ± 5.8	67.6 ± 5.6	103.7 ± 12.8	176.0 ± 32.4	147.3 ± 15.9	153.4 ± 24.5	149.1 ± 11.9	148.8 ± 26.9	176.2 ± 60.2	150.2 ± 6.8	163.6 ± 23.4	141.0 ± 7.7	69.2 ± 20.2	192.5 87.5±	137.1 ± 27.1	110.6 ± 11.9	130.4 ± 8.2	62.7 ± 12.0	51.2 ± 3.5	52.4 ± 4.6	51.8 ± 4.4	55.3 ± 4.8
Serine	80.9 ± 10.2	98.5 ± 12.6+	92.9 ± 14.7	82.9 ± 11.8	88.3 ± 15.2	98.6 ± 4.9	127.6 ± 18.7 ^a^	122.6 ± 6.5 ^a^	126.6 ± 15.3 ^a^	106.6 ± 5.3	161.4 ± 30.5	154.1 ± 49.6	143.6 ± 11.4	161.6 ± 26.8	130.3 ± 19.1	80.4 ± 8.0	201.1 ± 80.5	162.4 ± 49.8	141.6 ± 19.2	132.2 ± 11.2	101.2 ± 29.3	66.2 ± 8.3	61.9 ± 4.5	62.7 ± 11.0	56.2 ± 4.1
Tyrosine	60.2 ± 8.0	63.2 ± 7.0	67.3 ± 11.4	56.2 ± 8.7	53.4 ± 7.9	119.0 ± 20.9	231.5 ± 44.6 ^a^	192.6 ± 31.9 ^a^	209.9 ± 16.4 ^a^	176.1 ± 11.0 ^a^	130.6 ± 44.6	265.8 ± 63.7	189.6 ± 32.8	196.7 ± 19.2	175.2 ± 12.0	62.7 ± 12.1	212.8 ± 134.0	152.5 ± 37.5	94.5 ± 1.4	145.1 ± 45.2	66.2 ± 13.7	62.2 ± 14.0	62.6 ± 8.1	57.8 ± 6.3	76.9 ± 11.2

Data is presented as mean ±SEM, n = 5. ^a^
*p* < 0.05 vs. saline, ^b^
*p* < 0.05 vs. whey, ^c^
*p* < 0.10 vs. Vegan Mix. ^d^
*p* < 0.10 vs. Vegan Mix++. Vegan mix++ annotates the leucine-matched (to whey) vegan mix.

## Data Availability

Not applicable.

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
