# Peer review of "Dampened Muscle mTORC1 Response Following Ingestion of High-Quality Plant-Based Protein and Insect Protein Compared to Whey"

_nutrients, 2021, doi:10.3390/nu13051396_

Round 1

Reviewer 1 Report

The authors compared different protein sources for the amino acid availability in plasma and activation of mechanistic target of rapamycin complex 1 (mTORC1) in muscle. The potential reason(s) for the increase of amino acids in plasma in saline groups need to be discussed. The reason why the study is important, and how current conclusion link to the real world should be intensively mentioned.

Author Response

The authors compared different protein sources for the amino acid availability in plasma and activation of mechanistic target of rapamycin complex 1 (mTORC1) in muscle. The potential reason(s) for the increase of amino acids in plasma in saline groups need to be discussed. The reason why the study is important, and how current conclusion link to the real world should be intensively mentioned.

We thank the reviewer for his/her feedback.

Concerning the increase of plasma amino acids in the saline group: We carefully went through the data again but could not observe statistically significant differences in amino acids upon saline gavage (except for two small effects at 30min for glycine and cysteine, which disappeared at 60min). Obviously, we can speculate what might cause this 'increase': measurements errors, hour to hour variation and even a short-term stress response to the handling of the mouse. Another reason for the increased plasma AA 30-60 min after saline gavage could be alternations in amino acid turnover following circadian rhythms. A phenomenon that has been documented several decades ago by for instance Feigin et al, Am Journal of clin nutr 1971. Due to the lack of statistical significance, we feel it would not be appropriate to speculate about potential reasons for a non-existing difference.

We believe we extensively laid-out the importance of our study in line 40-49 (introduction) where we indicate that the demand for alternative protein sources is rapidly rising given the rise in world population and decreased willingness to consume animal-based protein sources.

We added a piece mentioning the clinical relevance of our data, with important cautionary points included as well. Line 274-283 in the discussion:

“The clinical relevance of our data lies in the nutritional support for alternative protein sources to activate mTORC1 and potentially muscle remodeling. Although the doses used in this study are similar to what is generally used in human studies after accounting for dose translation from rodent to humans [50], caution must be exercised to directly extrapolate our findings to humans. Mice and humans show differential digestion patterns. While mice have an exponential digestion curve, where most of the gastric emptying is finished within the first hour after nutrient ingestion, humans show a more linear digestion pattern [51]. This is a limitation of our study as interspecies digestion patters might have affected blood leucine concentrations at later time-points (e.g. 150 min) after gavage”

Reviewer 2 Report

Thank you very much for a well-written paper. 

Maybe you could emphasize the clinical importance of the paper more. 

Author Response

Thank you very much for a well-written paper. Maybe you could emphasize the clinical importance of the paper more. 

We thank the reviewer for his/her comment. We included a part mentioning the clinical relevance of our paper at the end of the discussion, with cautionary points in terms of extrapolation included as well

Line 274-283: “The clinical relevance of our data lies in the nutritional support for alternative protein sources to activate mTORC1 and potentially muscle remodeling. Although the doses used in this study are similar to what is generally used in human studies after accounting for dose translation from rodent to humans [50], caution must be exercised to directly extrapolate our findings to humans. Mice and humans show differential digestion patterns. While mice have an exponential digestion curve, where most of the gastric emptying is finished within the first hour after nutrient ingestion, humans show a more linear digestion pattern [51]. This is a limitation of our study as interspecies digestion patters might have affected blood leucine concentrations at later time-points (e.g. 150 min) after gavage.”

Reviewer 3 Report

Introduction (line 27 through 69) presents a logical and cogent series of arguments in support of the proposed study. The subject area is topical and impactful.

Line 70 hypothesise that whey protein would promote superior blood concentrations of EAA and branched-chain amino acids (BCAA) compared to total protein-matched vegan proteins or insect isolates, mirrored by superior acute mTORC1 activation. Additionally, we hypothesise to observe a similar EAA and mTORC1 response to whey when a vegan protein mix is leucine-matched to whey… is well constructed and can be challenged by an appropriate experimental design.

Line 93 Concentrations based on a nutritional supplement (containing 20 g whey protein and 2.8 g total leucine per serving used in humans to study its effect on muscle protein synthesis….. is difficult to fathom. 20g of protein and 2.8g in human studies equates to ~ 0.3g and 25mg per kg of protein and leucine, respectively. This is different from the stated 4.57g/kg protein and 475mg/kg (for Whey) used in this study on mice. Please explain.

Table 1:

  • How was protein value determined? Please state N:P value used to determine protein values for Whey, Vegan Mix and Worm proteins.
  • Aspartane or aspartate? Glutamine or glutamate? Tryptophane or tryptophan?
  • Why does sum AA/100g = 92.7 for worm?
  • If the amount gavaged is based on total protein, this equates to 5.7, 5.9, 7.4 and 7.6g of mix for whey, vegan mix, worm and vegan mix++, respectively, a difference in mass of 1.9g between the lowest mass 5.7g and highest 7.6g (i.e. 33%) between treatments. Would this influence digestion and subsequent appearance of amino acids in the bloodstream?
  • Similarly, the energy of each mix is also highly variable ranging from 333kcal/100g for Vegan mix to 498kcal/100g for worm (a 50% difference calculated from values presented in Table 1). Would this influence digestion and subsequent appearance of amino acids in the bloodstream?

Methods (sections 2.3 to 2.5 are appropriate and well presented)

2.6 Data analysis line 134 through 140.

  1. Please indicate test for normality of data prior to use of parametric stats
  2. Provide statistical power calculation for difference between treatments for n=5
  • Figure 2 presents data with N=5, (presumably derived from Table 2, which does state a value for N), yet the blots (Figure 3) have a value of N = 6-7 and Figure 4 N= 5-6?
  1. Please indicate method for calculating AUC (also see comment below)
  2. Results (line 141 …..)

Is AUC calculated setting the lower value to zero or the value at zero time point for each analyte?

Are the data presented as the AUC of the mean data at each time point or the mean of the individual AUCs?

The data for the saline (control) appears to show post-lavage temporal increase in plasma [AA]. Is this normal in mice to saline gavage?

Are fasted C57BL/6 mice gavaged with saline acting as the (null) control?  

If so, would it not be appropriate to determine the difference between the treatment and control values for each treatment? This would apply to both plasma AA and intracellular signalling.

Similar to the comment above, is the interpretation of these data by difference between the means or the mean of the differences?

  1. Discussion (line 195…..)
  2. Line 196 This study assessed AA disposal in the blood…there was no measurement of rate of disposal, only the temporal change in amino acid concentration
  3. Line 200 ‘whey induced a 2-fold larger increase in plasma BCAA 200 compared to plant-and insect-based proteins, with a subsequent 1.5-fold larger increase in downstream mTORC1 signalling’. Please revise and be more specific. Based on what parameter(s) of these data ([ ], AUC etc.) and compared to what (saline or between treatments)? This comment applies throughout the discussion.
  • Line 206 through….. This section refers to human studies. The authors have yet to declare whether experimental studies of this nature conducted in mice transfer directly to studies in humans. The assumption being that the process of digestion, absorption and release into the circulation does not differ between these two species. It would be pertinent to this discussion that a statement regarding the interspecies (note; also comparison to rats in line 244) similarity(ies)/difference(s) in these processes precedes this discussion.
  1. Line 234..BCAAs and in particular leucine,,,,this section refers to BCAA and/or leucine interchangeably yet, as recognised by the authors the principal extracellular AA regulator of MPS is Leucine, and much of the following discussion relates to leucine matched, not BCAA matched studies – this appears to be the authors’ preference when comparing between their data to the extant literature. This might warrant a statement relating to the differential size of effect on MPS to increasing [Leu], which is a single component vs. [BCAA] which can vary by protein source without a change in Leu?
  2. Line 240 Despite lower leucine availability… availability and concentration are not interchangeable, the recommendation is to continue with a concentration dependent discussion
  3. Line264 …The debate regarding mTORC1 ‘sensitivity’ to extracellular [leucine], with/without effect of other micro/macronutrients is considered as a comparative ‘dampening’ of activation and/or change in downstream protein kinases. This is a generalised and rather speculative discussion.

Line 266 Finally, further methods to increase digestion and absorption of plant-based proteins should be explored. Not sure how this emerges as a recommendation from the outcome of this study?

Author Response

Introduction (line 27 through 69) presents a logical and cogent series of arguments in support of the proposed study. The subject area is topical and impactful.

We thank the reviewer for his/her positive feedback on our work.

Line 70 hypothesise that whey protein would promote superior blood concentrations of EAA and branched-chain amino acids (BCAA) compared to total protein-matched vegan proteins or insect isolates, mirrored by superior acute mTORC1 activation. Additionally, we hypothesise to observe a similar EAA and mTORC1 response to whey when a vegan protein mix is leucine-matched to whey… is well constructed and can be challenged by an appropriate experimental design.

Thank you for acknowledging the importance of our research and the significance of our hypothesis.

Line 93 Concentrations based on a nutritional supplement (containing 20 g whey protein and 2.8 g total leucine per serving used in humans to study its effect on muscle protein synthesis….. is difficult to fathom. 20g of protein and 2.8g in human studies equates to ~ 0.3g and 25mg per kg of protein and leucine, respectively. This is different from the stated 4.57g/kg protein and 475mg/kg (for Whey) used in this study on mice. Please explain.

The reviewer is right. We deleted the referral to human studies and explained to how much of the total daily leucine intake our administered dose would correspond to. We believe this is a more appropriate way to frame our administration dose:

Line 93-95: “The concentrations chosen correspond to ~40% of total daily leucine intake and have been shown to induce a robust increase in mTORC1 signaling and MPS in rodents[29,33].”

That said, the referral to human doses as initially stated in the manuscript is correct when accounted for dose translation from human to rodents (Reagan-Shaw et al. FASEB, 2008). We elaborated on the extrapolation of our data to human at the end of the discussion:

Line 315-322 “Although the doses used in this study are similar to what is generally used in human studies after accounting for dose translation from rodent to humans [47], caution must be exercised to directly extrapolate our findings to humans. Mice and humans show differential digestion patterns. While mice have an exponential digestion curve, where most of the gastric emptying is finished within the first hour after nutrient ingestion, humans show a more linear digestion pattern[48]. We acknowledge that this is a limitation of our study as interspecies digestion patters might have affected blood leucine concentrations at later time-points (e.g. 150 min) after gavage.”

Table 1:

  • How was protein value determined? Please state N:P value used to determine protein values for Whey, Vegan Mix and Worm proteins.

The amino acid concentration for both whey, the vegan mix and worm protein was performed by internal assessments via the companies (Myprotein, Nutriprot and Protifarm respectively). We did not specifically measure N:P ratio for each mixture but added an estimation based on previous literature in the methods section:

Line 97-100: “AA content of each amino acid mixture was internally assessed by the respective company where the protein source was obtained. Nitrogen to protein ratio (N:P) for whey, vegan mix and worm protein is 6.38, 5.17-5.44 and 5.41 respectively. Values were obtained from previous studies [34–37].

  • Aspartane or aspartate? Glutamine or glutamate? Tryptophane or tryptophan?

Thank you for correcting this, we adapted the names of amino acids in Table 1

  • Why does sum AA/100g = 92.7 for worm?

We thank the reviewer for spotting this. We made a calculation error converting the values from ‘AA per dry weight powder’ to ‘fraction of each AA per 100g AA’. We corrected the mistake and added the correct values in Table 1.

  • If the amount gavaged is based on total protein, this equates to 5.7, 5.9, 7.4 and 7.6g of mix for whey, vegan mix, worm and vegan mix++, respectively, a difference in mass of 1.9g between the lowest mass 5.7g and highest 7.6g (i.e. 33%) between treatments. Would this influence digestion and subsequent appearance of amino acids in the bloodstream?

To be able to gavage the same (whey, vegan mix and worm) or more protein (but leucine-matched, vegan mix++) we indeed had to dissolve more vegan mix, worm and vegan ++ powder per volume water. By doing this, we kept the gavaged total volume constant between conditions. When rodents are administered pure leucine in different doses, mTORC1 response corresponds in a linear fashion to the administered dose of leucine, indicating that gastric emptying is not affected by high doses of leucine per volume water (Crozier et al, J Nutrition, 2005). Although we did not measure gastric emptying in our study, we assume that other anutritious factors might have affected AA disposal in the bloodstream. We mention this extensively in the discussion, line 224-232 and 242-246.

  • Similarly, the energy of each mix is also highly variable ranging from 333kcal/100g for Vegan mix to 498kcal/100g for worm (a 50% difference calculated from values presented in Table 1). Would this influence digestion and subsequent appearance of amino acids in the bloodstream?

Thank you for mentioning this valid point. Yes, other macronutrients might have affected gastric emptying and digestion. We added this point to the discussion.

Line 242-246: “Similarly, as shown in Table 1, the worm protein mixture contains more carbohydrates and fats than both the vegan mix and whey. The presence of other macronutrients, in particular fats, might have delayed gastric emptying and AA absorption in our study [34]. Future studies should investigate whether an insect source matched for leucine to whey and stripped from excess fats induces a similar leucine response.”

And

Line 278-282: “Mice and humans show differential digestion patterns. While mice have an exponen-tial digestion curve, where most of the gastric emptying is finished within the first hour after nutrient ingestion, humans show a more linear digestion pattern[51]. This is a limitation of our study as interspecies digestion patters might have affected blood leucine concentrations at later time-points (e.g. 150 min) after gavage.”

Methods (sections 2.3 to 2.5 are appropriate and well presented)

Thank you

2.6 Data analysis line 134 through 140.

  1. Please indicate test for normality of data prior to use of parametric stats

We have additionally performed Shapiro-Wilk tests for normality on the data presented in table 2, figure 2, figure 3 and 4. All data passed normality. We have added this to the methods section. (see point 2 below).

  1. Provide statistical power calculation for difference between treatments for n=5

We did a power calculation using G*Power software. Using a cohen (d) effect size of 1.5) we calculated that 5-7 mice per group is sufficient to reach a power of 0.8. We implemented such high effect sizes based on our previous work (D’Hulst et al, Nat Comm, 2020) and others (Crozier et al, J Nutrition, 2005) showing several fold increases (and hence large effect sizes) in phosphorylation of downstream mTORC1 kinases with similar doses of leucine than applied in this study.

We adapted the methods section according to the points the reviewer made.

Line: 145-148: “All data passed the Shapiro-Wilk test for normality and sample sizes (n = 5-7) were calculated a priori via power calculations (1-β: 0.8) using G*Power statistical software. Significance was set at p < 0.05. Exact n numbers are provided in the figure and table legends.”

  • Figure 2 presents data with N=5, (presumably derived from Table 2, which does state a value for N), yet the blots (Figure 3) have a value of N = 6-7 and Figure 4 N= 5-6?

Yes that is correct. The data presented in table 2 (figure 2), figure 3 and figure 4 are all generated from separate experiments using different mice (and n numbers).  We indicated the n number in table 2.

  1. Please indicate method for calculating AUC (also see comment below)
  2. Results (line 141 …..)

Is AUC calculated setting the lower value to zero or the value at zero time point for each analyte?

The AUC was calculated by subtracting the second value (e.g. leucine at time point 30) from the first value (e.g. leucine at time point 0). This value was multiplied by 30 (the time in minutes between time point 1 and 2) and divided by two. This was repeated for each time point and summated. This provides an AUC for each individual mouse of which then an average was plotted.

We added an explanation to the methods section:

Line 138-139: “Area Under the Curve was calculated using the linear trapezoidal method.”

Are the data presented as the AUC of the mean data at each time point or the mean of the individual AUCs?

The mean of the individual AUCs

The data for the saline (control) appears to show post-lavage temporal increase in plasma [AA]. Is this normal in mice to saline gavage?

We have not seen other papers reporting this. It might be normal daily variation in plasma amino acids following circadian rhythms. A phenomenon that has been documented several decades ago by for instance Feigin et al, Am Journal of clin nutr 1971. Moreover, the small rise in BCAA in the saline condition 30-60 min after gavage is not statistically significant, so it is difficult to make a strong, substantiated argument about this in the manuscript. The reviewer is referred to our answer to reviewer 1 who made a similar comment.

Are fasted C57BL/6 mice gavaged with saline acting as the (null) control?  

Yes, we opted to use saline gavaged mice as a 'sham' control treatment the control to account for mouse handling, the gastric filling and potential downstream hormonal effects of gastric stretching, etc.

If so, would it not be appropriate to determine the difference between the treatment and control values for each treatment? This would apply to both plasma AA and intracellular signalling.

We understand the reviewer's point of view. We however preferred to show the measured values rather than the differences between treatment and saline conditions.

Similar to the comment above, is the interpretation of these data by difference between the means or the mean of the differences?

We interpreted this data a being the difference between the means of each condition (saline vs whey vs vegan mix vs worm) statistically analysed via anova’s. The western blot data is semi-quantitative. This means that one can only interpret this data as being a ‘fold-change’ from one condition to another. For instance, pS6K1 was x-fold higher than saline and x-fold higher than worm. We cannot quantify the actual amount of phosphorylated proteins within each condition.

  1. Discussion (line 195…..)
  2. Line 196 This study assessed AA disposal in the blood…there was no measurement of rate of disposal, only the temporal change in amino acid concentration

    We agree with this comment and thank the reviewer for mentioning this.

Accordingly we changed this sentence in the discussion: Line 204-207: “This study assessed changes in AA disposal concentrations in the blood and mTORC1 response after gavage of animal-based proteins (whey), plant-based protein (a pea-rice vegan mix) and insect proteins (pulverized worm).”

  1. Line 200 ‘whey induced a 2-fold larger increase in plasma BCAA 200 compared to plant-and insect-based proteins, with a subsequent 1.5-fold larger increase in downstream mTORC1 signalling’. Please revise and be more specific. Based on what parameter(s) of these data ([ ], AUC etc.) and compared to what (saline or between treatments)? This comment applies throughout the discussion.

As this is the beginning of the discussion where we summarize our main findings, we would like not to go into too much detail on the specifics of mTORC1 signalling. We mean that whey induced a 1.5 fold larger increase in pS6k1 and pS6 compared to vegan mix and worm. To clarify this we added a section further down in the discussion:

Line 260-263: “Our data supports this as both vegan mixes induced lower rise in BCAA, which resulted in a 1.5-fold lower increase in phosphorylation of various downstream mTORC1 kinases such as p-S6K1 and p-S6 compared to whey”.

  • Line 206 through….. This section refers to human studies. The authors have yet to declare whether experimental studies of this nature conducted in mice transfer directly to studies in humans. The assumption being that the process of digestion, absorption and release into the circulation does not differ between these two species. It would be pertinent to this discussion that a statement regarding the interspecies (note; also comparison to rats in line 244) similarity(ies)/difference(s) in these processes precedes this discussion.

We agree that no comments about potential interspecies differences have been made. Therefore we added a section at the end of the discussion specifically addressing that:

Line 274-282: “The clinical relevance of our data lies in the nutritional support for alternative protein sources to activate mTORC1 and potentially muscle remodeling. Although the doses used in this study are similar to what is generally used in human studies after accounting for dose translation from rodent to humans [50], caution must be exercised to directly extrapolate our findings to humans. Mice and humans show differential digestion patterns. While mice have an exponential digestion curve, where most of the gastric emptying is finished within the first hour after nutrient ingestion, humans show a more linear digestion pattern [51]. This is a limitation of our study as interspecies digestion patters might have affected blood leucine concentrations at later time-points (e.g. 150 min) after gavage.”

  1. Line 234..BCAAs and in particular leucine,,,,this section refers to BCAA and/or leucine interchangeably yet, as recognised by the authors the principal extracellular AA regulator of MPS is Leucine, and much of the following discussion relates to leucine matched, not BCAA matched studies – this appears to be the authors’ preference when comparing between their data to the extant literature. This might warrant a statement relating to the differential size of effect on MPS to increasing [Leu], which is a single component vs. [BCAA] which can vary by protein source without a change in Leu?

Thank you for this valuable comment. We are aware of studies manipulating essential and branched-chain amino acid content while maintaining the same dose of leucine (Moberg et al, Am J Phys, Cell Phy, 2016) or studies that manipulated leucine content in different (non)-essential amino acid mixtures (e.g. Churchward-venne et al, Am J Clin Nutr 2014). As such, we are aware that elevations in valine and/or isoleucine might slightly increase muscle mTORC1 response to leucine, especially after exercise (Moberg et al, Am J Phys, Cell Phy, 2016). In our study, valine and isoleucine mirrored leucine concentrations independent of the amino acid mixture we gavaged. When plasma leucine was high, other BCAAs were higher. Similarly, when leucine concentration were dampened, other BCAAs were dampened (vegan, worm). Hence we believe discussing how differences in other BCAA concentrations (valine isoleucine) would affect mTORC1 independent of leucine, would not be relevant given our data set-up and aim of the study. It might only confuse the reader and draw the attention away from the main message of the paper.

If the reviewer agrees we would like to keep the focus on leucine as being the main driver of mTORC1 stimulation, when other amino acids are co-ingested.

  1. Line 240 Despite lower leucine availability… availability and concentration are not interchangeable, the recommendation is to continue with a concentration dependent discussion

We agree that amino availability and amino acid uptake into the muscle can be modulated by different factors (e.g. resistance training, Biolo et al, AJP endo, 1995), hence concentrations of leucine in the blood and what becomes readily available to tissues is not exactly the same. We adapted the discussion to focus our wording more on ‘concentration’ instead of ‘availability’.

  1. Line264 …The debate regarding mTORC1 ‘sensitivity’ to extracellular [leucine], with/without effect of other micro/macronutrients is considered as a comparative ‘dampening’ of activation and/or change in downstream protein kinases. This is a generalised and rather speculative discussion.

Thank you for this comment. The mere availability of other amino acids and/or growth factors have been shown to modulate mTORC1 signaling in response to leucine (Laplante & Sabatini, Cell, 2012,  Moberg et al, Am J Phys, Cell Phy, 2016, Kim et al, International J of Mol Med, 2015), so we believe that it is not far-fetched to state that other components in worm protein that are not available in plant and/or whey protein sensitize mTORC1 to leucine. If the reviewer agrees, we would like to keep this sentence in the manuscript as it evokes background for future research.

Line 266 Finally, further methods to increase digestion and absorption of plant-based proteins should be explored. Not sure how this emerges as a recommendation from the outcome of this study?

Thank you for this comment. We agree this might have been an overstatement, therefore we tuned-down our statement:

Line 292-295: “Finally, further methods to increase the ability of plant-based proteins to activate muscle protein synthesis should be explored: these might include improved means to increase di-gestion and absorption”

Reviewer 4 Report

The authors take up an extremely important topic, the article is very interesting and valuable.

Have the authors considered undertaking a histological analysis of the tibialis anterior muscle in terms of the impact of high-quality plant-based protein and insect protein compared to whey on muscle hypertophy or hyperplasia by determining e.g. the diameter of muscle fibers or their number? 

Author Response

The authors take up an extremely important topic, the article is very interesting and valuable.

Have the authors considered undertaking a histological analysis of the tibialis anterior muscle in terms of the impact of high-quality plant-based protein and insect protein compared to whey on muscle hypertophy or hyperplasia by determining e.g. the diameter of muscle fibers or their number? 

We thank the reviewer for his/her positive feedback on our manuscript. As this is an acute study without any long-term resistance exercise intervention, it seemed very unlikely to us that the effects we see from the different protein sources on mTORC1 would transfer to acute alternations on fiber hypertrophy.

To the reviewer’s point. More research is needed to pinpoint the effects of plant-based, and certainly insect-based protein sources, on long-term training adaptions in rodents and humans. We hope our acute study forms the backbone for such experiments.

Round 2

Reviewer 1 Report

The authors have answered all questions raised by the reviewer.

Reviewer 3 Report

The response to the reviewer's comments and revisions to the 1st draft of this m/s are acceptable. The author's acknowledge the rather large list of 'errors' in the 1st draft. Future submissions to Nutrients should be prepared better.

There remain diagreements between the reviewer and the author, but that is the nature of peer review.

The revised m/s has benefitted from peer review.